# Can Diffusion Models Generalize? Privacy and Fairness Trade-offs for Medical Data Sharing.

**Mischa Dombrowski**[1]             MISCHA.DOMBROWSKI@FAU.DE
**Bernhard Kainz**[1,2]              BERNHARD.KAINZ@FAU.DE

[1] *Friedrich-Alexander-Universität Erlangen-Nürnberg, DE*

[2] *Imperial College London, UK*

**Editors:** Accepted for publication at MIDL 2025

## Abstract

The recent surge in options for diffusion model-based synthetic data sharing offers significant benefits for medical research, provided privacy and fairness concerns are addressed. Generative models risk memorizing sensitive training samples, potentially exposing identifiable information. Simultaneously, underrepresented features – such as rare diseases, uncommon medical devices, or infrequent patient ethnicities – are often not learned well, creating unfair biases in downstream applications. Our work unifies these challenges by leveraging artificially generated fingerprints (SAFs) in the training data as a controllable test for memorization and fairness. Specifically, we measure whether a diffusion model reproduces these fingerprints verbatim (a privacy breach) or ignores them entirely (a fairness violation) and introduce an indicator t' to quantify trained models for the likelihood of reproducing training samples. Extensive experiments on real and synthetic medical imaging datasets reveal that naïve diffusion model training can lead to privacy leaks or unfair coverage. By systematically incorporating SAFs and monitoring t', we demonstrate how to balance privacy and fairness objectives. Our evaluation framework provides actionable guidance for designing generative models that preserve patient anonymity without excluding underrepresented patient subgroups. Code is available at https://github.com/MischaD/Privacy.

**Keywords:** Image Generation, Privacy, Fairness, Chest X-ray

## 1. Introduction

Since the development of statistical models capable of representing and synthesizing new samples from existing dataset distributions (Kingma and Welling, 2014; Goodfellow et al., 2020; Rombach et al., 2022; Ho et al., 2020; Hamamci et al., 2025; Guo et al., 2024), the idea of training generative models on private data and sharing *only* the model or synthetic datasets has gained traction. Such methods could address issues like data scarcity for rare diseases, racial bias (Larrazabal et al., 2020), and challenges such as robust domain adaptation and generalisation (Wang et al., 2022). However, maintaining privacy and anonymity is crucial when working with personally identifiable information (Jin et al., 2019).

Recent advances in generative modeling, including diffusion models (Song et al., 2020a; Dhariwal and Nichol, 2021; Rombach et al., 2022; Ruiz et al., 2022), have expanded the feasibility of sharing models directly (Pinaya et al., 2022). Despite these efforts (Bai et al., 2021; Dar et al., 2024; Stein et al., 2024), it remains unclear to what extent shared models reproduce training samples, which would raise potential data privacy concerns. Guarantees against such privacy breaches would allow models to be trained on proprietary data and shared instead of the underlying datasets, enabling fully anonymous data sharing.

However, recent works have shown that publicly released models can inadvertently regenerate training data during sampling, which prevents us from freely sharing such models. For instance, Somepalli et al. (2023) and Dar et al. (2024) demonstrate that diffusion models can reproduce training samples, while Carlini et al. (2023) illustrate how re-identifyable faces can be extracted from these datasets. This raises serious privacy concerns requiring robust mitigation strategies (Ren et al., 2024). Moreover, some generative models are explicitly designed to memorize training samples (Cong et al., 2020). Differential privacy-based strategies (Dockhorn et al., 2022) offer a promising avenue for mitigating these concerns. However, their adoption in high-resolution data synthesis remains limited due to a notable drop in distribution fidelity, restricted applicability to diffusion models, and low efficacy for multi-modal, high-resolution datasets (Xie et al., 2018). Consequently, in this work, we propose a direct approach to evaluate models – circumventing the need to obfuscate the model distribution – and thereby preserving compatibility with high-resolution, high-fidelity generative tasks.

Additionally, not reproducing unique features has important implications for the fairness of generated data, which have not yet been discussed in literature. Training diffusion models on private datasets while ensuring unique features are not reproduced results in models ignore these unique characteristics. This contradicts the objective of generating a fair dataset, where such unique features should be reproduced. To resolve the trade-off between privacy and fairness, models must learn to generalize. To approach this problem, we can conceptually divide the set of all images into two distinct categories: training set members and non-members. After generating a dataset using a diffusion model, the generated samples fall into four categories: (1) **Lost images**: not in the training set, unavailable, potentially affecting downstream tasks but posing no privacy or fairness concerns. (2) **Memorized images**: reproduced from the private training set, raising privacy issues that require safeguards. (3) **Forgotten images**: training images not generated, potentially causing fairness issues if certain subgroups are omitted. (4) **Generalized samples**: the ideal case, where the model generates data reflecting the underlying distribution.

To investigate what the model learns, we propose to use synthetic anatomical fingerprints. These fingerprints can be directly controlled through synthetic manipulations of the training dataset and reliably detected in synthetic datasets. We measure the probability of generating these fingerprints using a novel indicator metric $t'$. Thus, our main contributions are:

- We formulate a realistic scenario where unconditional generative models face privacy and fairness problems due to the potential reproduction of training samples.
- We provide a formal approach to determine the maximum probability of producing sensitive data, from which we derive a computable indicator metric.
- We define a framework that quantifies and investigates privacy and fairness issues, enabling architectural decisions to create truly generalizing and fair generative models.

## 2. Background

**Diffusion Models**, such as (Rombach et al., 2022), model different levels of perturbation $p_\sigma(\tilde{\mathbf{x}}) := \int p_{data}(\mathbf{x}) p_\sigma(\tilde{\mathbf{x}} \mid \mathbf{x}) d\mathbf{x}$ of the real data distribution using a noising function defined by $p_\sigma(\tilde{\mathbf{x}} \mid \mathbf{x}) := \mathcal{N}(\tilde{\mathbf{x}}; \mathbf{x}, \sigma^2 \mathbf{I})$. Here, $\sigma$ defines the strength of the perturbation, split into $N$ steps $\sigma_1, \ldots, \sigma_N$. The assumption is that $p_{\sigma_1}(\tilde{\mathbf{x}} \mid \mathbf{x}) \sim p_{data}(\mathbf{x})$ and $p_{\sigma_N}(\tilde{\mathbf{x}} \mid \mathbf{x}) \sim \mathcal{N}(\mathbf{x}; \mathbf{0}, \sigma_N^2 \mathbf{I})$. We

define the optimization as a score matching objective by training a model $\mathbf{s}_{\boldsymbol{\theta}}(\mathbf{x}, \sigma)$ to predict the score function $\nabla_{\mathbf{x}} \log p_{\sigma}(\mathbf{x})$ for the noise level $\sigma \in \{\sigma_i\}_{i=1}^{N}$. For sampling, this process can be reversed, for example, using Markov chain Monte Carlo methods following Song and Ermon (2019). Song et al. (2020b) extended this to a continuous formulation by redefining the diffusion process as a process governed by an SDE and training a dense model to predict the score function. The continuous formulation of the noising process, denoted by $p_t(\mathbf{x})$ and $p_{st}(\mathbf{x}(t) \mid \mathbf{x}(s))$, characterizes the transition kernel from $\mathbf{x}(s)$ to $\mathbf{x}(t)$, where $0 \leq s < t \leq T$. Anderson (1982) showed that the reverse of this diffusion process is also a diffusion process. Finally, Song et al. (2020b) show that the reverse diffusion process of the SDE can be modeled as a deterministic process, as the marginal probabilities can be expressed deterministically in terms of the score function. As a result, the problem simplifies to an ODE, which can be solved using any black-box numerical solver, such as the explicit Runge-Kutta method. This enables exact likelihood computation, commonly used to estimate the likelihood of generating a sample, *e.g.*, images (Song et al., 2020b). However, we propose $t'$, a more general indicator that extends this idea to approximate the likelihood of generating all samples that lead to privacy problems.

**Privacy:** To formalize and contextualize our approach, we borrow the definitions of *extractable memorization* and *discoverable memorization* from the natural language processing domain (Nasr et al., 2023; Carlini et al., 2021) and apply them to generative image models. Given a model $\mathbf{s}$ with a generation routine Gen, an example $\mathbf{x}_p$ from the training set $D$ is *extractably memorized* if an adversary (without access to $D$) can construct a conditioning $\mathbf{c}$ that makes the model produce $\mathbf{x}_p$ (*i.e.*, $\text{Gen}(\mathbf{c}) = \mathbf{x}_p$).

We also adopt and extend the definition of *discoverable memorization* from Nasr et al. (2023) and Carlini et al. (2021) to image models: For a model $\mathbf{s}$ with generation routine Gen, an example $\mathbf{x}_p \in D$, and a perturbation function from the generative model's training $p_{\sigma}(\tilde{\mathbf{x}} \mid \mathbf{x})$, $\mathbf{x}_p$ is *discoverably memorized* if $\text{Gen}(\tilde{\mathbf{x}}_p, \sigma) = \mathbf{x}_p$. The strength of the perturbation function directly influences how discoverable the training images are. Our proposed indicator $t'$ measures the susceptibility of models to discoverable memorization. It can be compared to the privacy budget in differential privacy (Dockhorn et al., 2022); however, unlike differential privacy methods – which only work on low-resolution images – our approach' post-hoc nature preserves image quality.

**Fairness:** Fairness in AI is a well-explored yet unsolved problem. Current directions in the literature for discriminative tasks suggest frameworks for benchmarking (Jin et al., 2024) or reveal important design choices for training fair models. Generative models are used to improve fairness (Ktena et al., 2024), but their own biases and unfairness remain underexplored. It is often assumed that generative models learn the entire data distribution without further evaluation.

## 3. Method

To ensure clarity, we first define the key terms used throughout our paper. In Fig. 1 we visualize the key terms introduced Sec. 1.

**Privacy**: Sharing synthetic data poses a problem if an adversary, without access to any images from the training dataset but with prior knowledge, can extract an image from the synthetic dataset and recognize that it is memorized. An example of this is shown in Sec. 4.

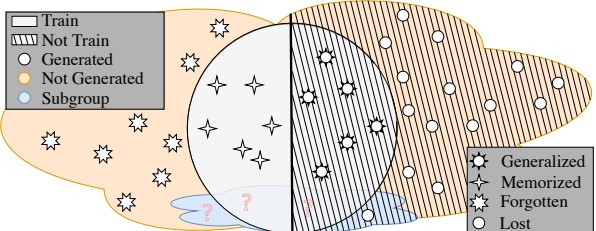

Figure 1: Type of learning for generative models. Memorization raises privacy issues, and forgetting raises fairness issues.

**Memorization**: Memorization refers to the pixel-wise reproduction of training images. We distinguish between the memorization of full images and parts of images. To check for partial memorization, we define fingerprints a priori. Instead of computing a pixel-wise error, we use a classifier $C_f$ that is trained to be robust against perturbations. The second type is full-image memorization, where the model reproduces the entire image.

**Identity**: This refers to case-dependent sensitive information that may be leaked if the model is shared. Examples include full images in the case of ChestX-ray14 (Wang et al., 2017) or CelebA-HQ (Karras et al., 2018). In other cases, such as the Stable Diffusion experiment in Sec. I, identity refers to the actual identity of a person, independent of image context or background.

**Fingerprint**: An image feature unique to a specific person or image in the dataset. Examples include distinctive diseases, objects, entire faces, bone structures, or any feature linked to an individual's identity. Similar to real fingerprints, their mere presence is not inherently problematic unless there is additional information that enables identification.

**Violation**: This occurs when revealing the identity is implied by the reproduction of a fingerprint. Formally, this means that the presence of a fingerprint implies the presence of the identity.

Using artificially generated fingerprints (SAFs), we investigate privacy and fairness issues simultaneously. Privacy issues arise if we detect cases of memorization. Fairness issues arise if SAFs are absent in synthetic datasets. Our goal is to achieve generalization, where SAFs are reproduced in images with different identities. To investigate when models start to generalize, we artificially inject detectable objects into the training data, *i.e.*, SAFs. We then train one classifier to detect these objects and another to identify the image's identity used as the target for injection. A non-privacy-violating and fair model would reproduce the SAF on a synthetic image with a different identity than the training image containing the fingerprint. To establish SAFs, we synthetically augment a single sample $\mathbf{x}_p$ from the dataset $D$. We do this too simplify the training of classifiers used to detect SAFs. In practice, this can be any feature that appears only once in the entire training dataset, such as a ring, a deformation, or a specific medical device. To investigate generalizability, we experiment with using either a constant gray circle, realistic fingerprints generated through image interpolation, or real features extracted based on labels as shown in Appx. E.

To assess whether the trained model raises privacy concerns, we define an adversarial attacker aware of the SAF who can train a model to detect it. We refer to this classifier as $C_f$. A second classifier, $C_{id}$, is trained independently of the fingerprint on the unaugmented dataset using a one-versus-all approach to classify the image's identity. The attacker does not have access to this classifier. Its purpose is to determine if the presence of the fingerprint implies the identity of the image.

This setup allows us to disentangle the memorization of the SAF from the memorization of $\mathbf{x}_p$, distinguishing generalization from memorization. To track the number of memorized samples, we define $|q|$ as the number of synthetic samples where both classifiers have a positive outcome.

**Memorization Indicator** $t'$**:** It is possible to compute the likelihood of the exact sample (*e.g.*, using numerical NLL estimation), but this does not ensure that images in the immediate neighborhood are free from privacy issues. To address this, we propose estimating the upper bound of the likelihood of reproducing samples from the entire subspace belonging to the class of private samples.

Let $p_s(\mathbf{x}_p)$ define the likelihood of the unconditional model $\mathbf{s}$ reproducing the private sample $\mathbf{x}_p$ at test time. This is insufficient because it does not account for slightly noisy versions of $\mathbf{x}_p$, which can also pose privacy concerns. We aim to compute $q(p)$, defined as the likelihood of reproducing any sample within $\Omega_p$. Here, $\Omega_p$ represents the region in image space that is similar enough to $\mathbf{x}_p$ to raise privacy concerns according to $q$.

In the supplementary material, we show this is equivalent to:

$$q(p) = \int_{\Omega_p} p_s(\mathbf{x})\mathrm{d}\mathbf{x} \approx \int_0^{t'} p_s(\mathbf{x}_{t,p})\mathrm{d}\mathbf{t} \le \sum_{i=0}^{t'} \sup_{t \in [t_i, t_{i+1}]} (\sigma_{t_{i+1}} - \sigma_{t_i}) \mathbb{E}_{p(\mathbf{x}_{t,p})}\big[p(\mathbf{x}'_{t,p})\big], \quad (1)$$

To estimate $q(p)$, we observe that it depends only on the likelihood $p(\mathbf{x}'_p)$ and $t'$, which captures the entire region of $\Omega_p$. $\mathbf{x}'_p$ is the predicted sample of the diffusion model after applying $t$ forward diffusion steps to the private sample $\mathbf{x}_p$ . This synthetic $\mathbf{x}'_p$ then serves as input to the classifiers. $\Omega_p$ is defined as the region where $C_{id}$ and $C_f$ both give positive predictions. Since this region depends only on its size, $t'$ serves as an indicator of how unlikely it is to generate critical samples from the model, without the necessity to compute the exact value for $p(\mathbf{x}'_p)$ .

We provide a pseudo-algorithm for the computation in Appx. G. Given $\mathbf{x}_p$, we define $q_M(p|x_{t,p})$ as the estimate of a sample belonging to $\Omega_p$ for a given diffusion step $t$. We then define $t' := \max(\mathbb{T})$, where $\mathbb{T} := \{\forall t : q_M(p|x_{t,p}) > 0\}$. The parameter $M$ allows us to trade off accuracy for computation time by choosing the number of generated samples.

**Intuition:** We model the image space using the learned distribution of the score function $\nabla_{\tilde{\mathbf{x}}} \log p_{\sigma_i}(\tilde{\mathbf{x}} \mid \mathbf{x})$ by reversing the diffusion process and checking when the model starts to "break out" by generating images classified as different samples. For large $t$, the learned marginals $p(\mathbf{x}, t)$ span the entire image space. Importantly, by definition of the diffusion process, the distribution approaches the same distribution as the sampling distribution of the diffusion process if $\sigma_t$ gets large enough $p_{\sigma_N}(\tilde{\mathbf{x}} \mid \mathbf{x}_p) \sim \mathcal{N}(\mathbf{x}; \mathbf{0}, \sigma_N^2 \mathbf{I})$. However, for lower $t$ the model has learned that the distribution collapses towards a single training image $\mathbf{x}_p$. Essentially, it has modeled part of the subspace as a delta distribution around $\mathbf{x}_p$. We want to estimate how far back in the diffusion process we have to go for the model to start to

produce different images. The boundary $\Omega_p$ is defined as all images that would collapse towards this training image, estimated using the classifiers. Fig. 6 illustrates this process in one dimension. The indicator t' is then the strength of the perturbation function according to the definition of discoverable memorization introduced in Sec. 2. Note that this is different from simply defining a variance that is large enough for the classifiers to fail, as $s_\theta(\mathbf{x}_p, \sigma_t)$ was trained to revert this noise. Fig. 7 illustrates how this looks in image space.

**Computational Overhead:** Our proposed method computes $t'$ through forward passes of the diffusion model, making its computational cost equivalent to that of image sampling. The hyperparameter $M$ determines the trade-off between the accuracy of $t'$ and computational overhead, scaling linearly with $M$. For instance, with $M = 16$, ensuring privacy for an image requires 16 times the computational cost of generating a single sample.

## 4. Experiments

We consider the size of the training dataset, the time for training, and model size as the three most impactful factors determining a model's fairness and memorization capabilities. **Dataset:** For our initial experiments we use an a-priori selected selection of modalities from MedMNISTv2 (Yang et al., 2021). For our main experiments, we use ChestX-ray14 (Wang et al., 2017), a dataset of 112,120 frontal chest X-rays widely studied in privacy research (Packhäuser et al., 2022). Additional experiments on training length and number of fingerprints per dataset, we use three datasets with diverse modalities and sizes. Specifically, we report results on BCI (Liu et al., 2022) and ODIR-2019 (https://odir2019.grand-challenge.org/dataset/). Data is split (60/20/20), with diffusion models sharing training data with classifiers.

**Metrics:** To evaluate generative quality, we report the Fréchet inception distance (FID). To gauge memorization, we compute the peak signal-to-noise ratio (PSNR) between 1000 synthetic samples and all images of the training sets, reporting the maximum value. To quantify privacy, we compute $t'$. To quantify fairness, we compute the number $|C_{f+}|$ of positive predictions.

**Toy Dataset:** We start by training $C_{id}$, $C_f$ and diffusion models on MedMNIST. Training details and comprehensive quantitative results can be found in Appx. H. Next, we compute t' by evaluating $q_M(p|x_{t,p})$. Examples of model input and output as well as a visualization of the evaluation process are shown in Fig. 7 and Fig. 3. In general we observe that t' is higher if the number of reproduced memorized samples is high. The cut-off value when the model stops to reproduce the SAF seems to be around $t' = 0.7$.

**Training Length:** To establish the generalizability of our approach, we extend our experiments to real datasets (ChestX-ray14, BCI,ODIR-2019). First, we experiment with the training length. We compute FID at different training lengths and select the best model based on the lowest value. For this we use PII (Tan et al., 2021) as our SAFs. The results are shown in Fig. 2. For ChestX-ray14, overtraining results in higher PSNR and FID values. The SAF itself is never reproduced. BCI has a few positive predictions during earlier epochs, which turn out to be false positives, likely caused by lower sample quality during early training. The best epoch, based on FID, does not reproduce the fingerprint at all. The high PSNR is caused by samples with large empty regions. ODIR-2019 shows a high number of reproduced SAF and a high PSNR, indicating memorization. Positive predictions confirm

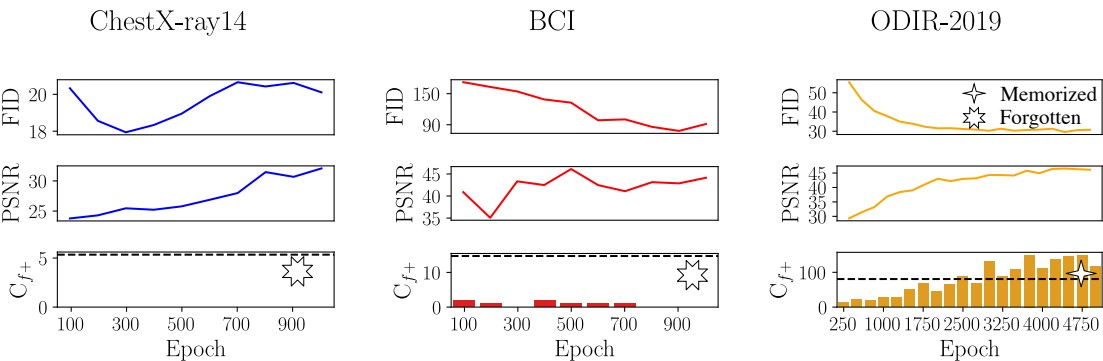

Figure 2: Impact of training length. To assess model memorization, we investigate PSNR and $C_{f+}$. The dashed line in the bottom row indicates the results of a fair model (SAFs appear equally often in training and synthetic datasets).

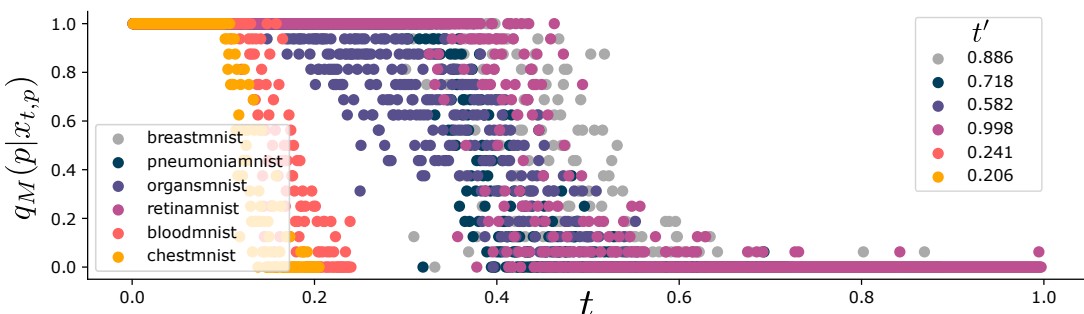

Figure 3: Likelihood of producing $\mathbf{x}_p$ at sampling time as a function of $t$ for $M = 16$.

that all are copies of the training sample. Overall, models either memorize the SAF, leading to privacy issues, or show limited sample diversity, resulting in unfair outcomes.

We repeat this experiment with the ideal training lengths and train diffusion models with different types of fingerprints to assess generalizability to real fingerprints. Results are provided in Appx. E. None of the ChestX-ray14 and BCI models reproduce the SAFs, indicating that none of the models are fair. The results for the three different SAFs differ slightly when examining $C_{f+}$ and $t'$. Generative models struggle with synthetic features like the circle, while PII shows intermediate retention due to its inpainting method.

**Dataset Size** To analyze the impact of dataset size, we use a latent diffusion model for its higher efficiency and sample quality, leveraging the VQ-VAE from Stable Diffusion 2.0 (Rombach et al., 2022). Details on training parameters are in Appx. B. The results are shown in Tab. 1. Interestingly, FID values are negatively correlated with dataset size, likely due to the model's ability to memorize training data, which produces realistic images. Inspecting these images reveals that all are memorized. For smaller datasets, the training image can often be retrieved from the sampled images, as indicated by the high $t'$ values.

Table 1: Quantitative results on CXR data using two backbones: OD (out-of-domain) Inception and ID (in-domain) models for CXR (Cohen et al., 2022). Larger datasets reduce memorization risk, quantified by $t'$.

|  | $|N_D|$ | Classification | | OD (Inception) | | ID (CXR) | | Privacy | | |
|---|---|---|---|---|---|---|---|---|---|---|
|  |  | SAF (%) | ID (%) | $\mathrm{FID_{train}}$ | $\mathrm{FID_{test}}$ | $\mathrm{FID_{train}}$ | $\mathrm{FID_{test}}$ | $\mathbb{E}(|q|)$ | $|q|$ | $t'$ |
| Chestxray14 | 875 | 100.00 | 100.00 | 15.1 | 30.3 | 1.0 | 2.3 | 34.3 | 47 | 0.75 |
|  | 1750 |  |  | 12.3 | 29.3 | 1.0 | 2.4 | 17.1 | 5 | 0.86 |
|  | 3500 |  |  | 13.6 | 32.0 | 1.2 | 2.6 | 8.6 | 1 | 0.67 |
|  | 7001 |  |  | 18.8 | 38.4 | 1.6 | 3.0 | 4.3 | 0 | 0.72 |
|  | 14003 |  |  | 22.1 | 41.4 | 1.9 | 3.3 | 2.1 | 0 | 0.66 |
|  | 28007 |  |  | 19.9 | 39.4 | 2.1 | 3.4 | 1.1 | 0 | 0.60 |

Table 2: Privacy metrics for different model sizes. See Appx. D for the different hyperparameters we use for the diffusion model.

|  | # Trainable parameters | $|q|$ | FID | $t'$ |
|---|---|---|---|---|
| Default | 113 675 524 | 5 | 32.7 | 0.77 |
| Model 1 | 77 364 740 | 9 | 33.0 | 0.74 |
| Model 2 | 71 439 108 | 0 | 33.9 | 0.69 |
| Model 3 | 49 558 020 | 1 | 34.8 | 0.69 |
| Model 4 | 28 484 612 | 0 | 78.7 | 0.66 |
| Model 5 | 28 448 388 | 0 | 43.6 | 0.64 |

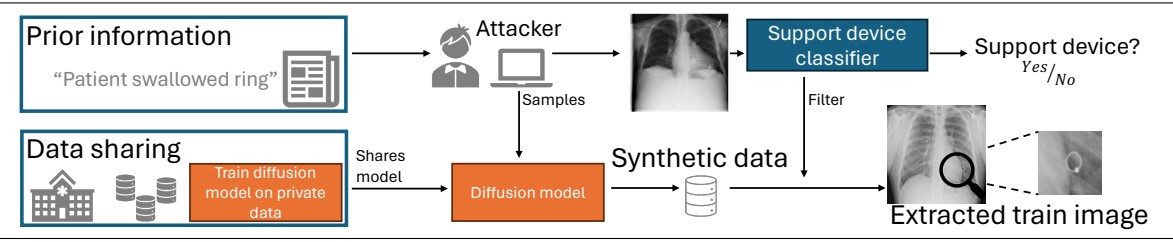

Figure 4: Extracting memorized samples from a trained diffusion model. The attacker learns that a ring is in the patient's image, uses this information, and filters generated samples until reproducing the training image.

Notably, while sampling with $|N_D| = 7001$ does not generate SAFs, the high $t' = 0.72$ suggests privacy risks. For larger datasets, $t'$ is lower, indicating better privacy. In all cases, the value of $|q|$ deviates from the expected one, highlighting fairness issues.

**Model size:** Ideally, we want a model that avoids memorizing the fingerprint without requiring more training images. To address this, we investigate how model architecture size influences memorization by training models with fewer parameters on $|N_D| = 1770$ images. Details about the architecture are in Appx. D. The results are shown in Tab. 2. Smaller models stop reproducing the image, as indicated by the decreasing $t'$, but this reduces quality, as shown by the worse FID values. Manual inspection reveals that fingerprints are either memorized or forgotten across all models.

**Real-world example:** Next we want to highlight the severity of this problem by presenting a realistic scenario where an attacker leverages prior knowledge to extract private data from diffusion models. We assume the attacker knows that a patient has swallowed, *e.g.*, a ring, a scenario observed in our dataset. The attacker, without access to $\mathbf{x}_p$, could train a classifier to detect objects other than soft tissue and use it to filter sampled images, as shown in Fig. 4. To test this, we manually label the presence of support devices (*e.g.*, pacemakers, tubes) in 5000 images from (Wang et al., 2017) and train $C_f$ to detect them. We generate a synthetic dataset using a diffusion model trained on a set without support devices except for one image containing a ring to simulate accidental inclusion. The resulting model has a high $t' = 0.82$.

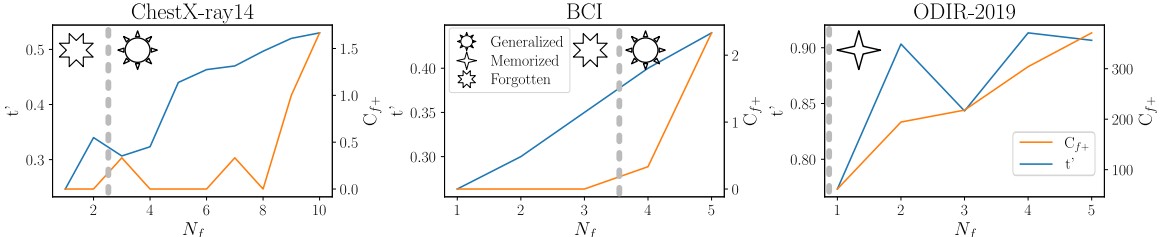

Figure 5: Number of fingerprints in the synthetic dataset and t' relative to $N_f$. The grey line marks when models begin reproducing fingerprints.

Using $C_f$, we filter 342 sampled images, most of which are false positives due to the difficulty of detecting unspecific support devices in medical images. Manual inspection confirms that only memorized training samples contain rings. This demonstrates how an attacker could use knowledge of a ring's presence in the training dataset to reconstruct the CXR image.

**Achieving Fairness:** To address the observation that all models raise fairness or privacy issues, we investigate the number of images $N_f$ containing the fingerprint as a key factor for generalization. The results are shown in Fig. 5. For ChestX-ray14, even with three fingerprinted samples, the model has a slight chance of reproducing the fingerprint, with one successful run out of three. At nine samples, the model begins reproducing the fingerprint more often, but the proportion remains low. Only one in 50,000 synthetic samples contains the fingerprint, even though 0.1% of the training data does. Generalization begins after roughly three samples. BCI generalizes after about four fingerprinted images, despite having a smaller training size. For ODIR-2019, the small dataset size causes memorization, showing that training diffusion models on small datasets does not enable generalization.

**Limitations:** Because the indicator $t'$ depends on the inherent stochasticity of sampling $q_M(p \mid x_{t,p})$, it may exhibit notable variance for larger values. As a result, outcomes with $t'$ close to 1 can be more challenging to compare and interpret. Moreover, although our framework provides robust privacy and fairness assurances at the sample level, extending these protections to entire datasets remains an open avenue for future work.

## 5. Conclusion

We described scenarios where training generative models on personally identifiable image data can lead to training data leaks. Using our framework, we successfully investigated common diffusion model design parameters. Our work reveals that, regardless of design choices, models are either not privacy-preserving or raise fairness issues by forgetting important long-tail information. Rare diseases will either be forgotten or memorized by the diffusion model. This must be considered and can be quantified by using $t'$ when designing models for data sharing and we observe that increasing the range of unique features in a dataset fosters improved generalization. The only alternative to avoiding memorization is to increase the size of the training dataset. However, in such cases, fairness should be carefully assessed, particularly concerning long-tail data.

## Acknowledgments

This work was supported by the High-Tech-Agenda Bavaria. HPC resources were provided by the Erlangen National High Performance Computing Center (NHR@FAU) of the Friedrich-Alexander-Universität Erlangen-Nürnberg (FAU) under the NHR project b143dc and b180dc. NHR funding is provided by federal and Bavarian state authorities. NHR@FAU hardware is partially funded by the German Research Foundation (DFG) – 440719683. Support was also received by the ERC - projects MIA-NORMAL 101083647 as well as DFG 513220538, 512819079 and DFG large scale infrastructure funding Art 91b GG.

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

## Appendix A. Derivation of Estimation Indicator t'

Song et al. (2020b) show that the reverse diffusion process of the SDE can be modeled as a deterministic process as the marginal probabilities can be modeled deterministically in terms of the score function. As a result, the problem of learning transition kernels simplifies to an ODE:

$$\mathrm{d}\mathbf{x} = \Big[\mathbf{f}(\mathbf{x},t) - \frac{1}{2}g(t)^2\nabla_{\mathbf{x}}\log p_t(\mathbf{x})\Big]\mathrm{d}t, \tag{2}$$

Solving Eqn. 2 enables exact likelihood computation. However, this does not account for the fact that images in the immediate neighborhood, like slightly noisy versions of $\mathbf{x}_p$, are not anonymous. Consequently, we are interested in computing $q(p)$, which is defined as the likelihood of reproducing any sample within $\Omega_p$, which is the region of the image space that is similar enough to $\mathbf{x}_p$ that it raises privacy concerns:

$$q(p) = \int_{\Omega_p} p_s(\mathbf{x})\mathrm{d}\mathbf{x}. \tag{3}$$

We determine this region by training a classifier tasked with detecting whether the image belongs to the image class $C_f$. To search through the image manifold, we make use of the reverse diffusion process centered around the SAF image $\mathbf{x}_p$ defined as $p_{t,b} \coloneqq p(\mathbf{x}_t \mid \mathbf{x}_p) = \mathcal{N}(\tilde{\mathbf{x}}; \mathbf{x}_p, \sigma_t^2\mathbf{I})$ for $\mathbf{x}(s)$ to $\mathbf{x}(t)$, where $0 \le t \le T$. We can employ the diffusion process centered around this image to sample from the neighborhood and then use the learned reverse diffusion process to generate noisy samples $\mathbf{x}_{t,p}$. Then we can use this as starting image for the reverse diffusion process to sample $\mathbf{x}'_{t,p}$:

$$q(p) = \int_{\Omega_p} p_s(\mathbf{x})\mathrm{d}\mathbf{x} \approx \int_0^{t'} p_s(\mathbf{x}_{t,p})\mathrm{d}\mathbf{t} = \int_0^{t'} \mathbb{E}_{p(\mathbf{x}_{t,p})}\big[p(\mathbf{x}'_{t,p})\big]\mathrm{d}\mathbf{t}. \tag{4}$$

Technically, we could employ exact likelihood computation to estimate $q(p)$ but this would require integrating over the continuous image-conditioned diffusion process, which would be intractable in practice. Therefore, we propose to approach and estimate this integral by computing the Riemann sum of this integral and give an upper bound estimate for it using the upper Darboux sum:

$$\int_0^{t'} \mathbb{E}_{p(\mathbf{x}_{t,p})}\big[p(\mathbf{x}'_{t,p})\big]\mathrm{d}\mathbf{t} =$$

$$\sum_t (\sigma_t - \sigma_{t-1})\mathbb{E}_{p(\mathbf{x}_{t,p})}\big[p(\mathbf{x}'_{t,p})\big] \le \sum_{i=0}^{t'} \sup_{t\in[t_i,t_{i+1}]} (\sigma_{t_{i+1}} - \sigma_{t_i})\mathbb{E}_{p(\mathbf{x}_{t,p})}\big[p(\mathbf{x}'_{t,p})\big], \tag{5}$$

which approaches the real value for steps that are small enough. We can compute this value by using $\mathbf{x}_p$ as a query image and estimating the expectation by performing Monte-Carlo sampling but this would be computationally infeasible due to the complexity of exact likelihood estimation.

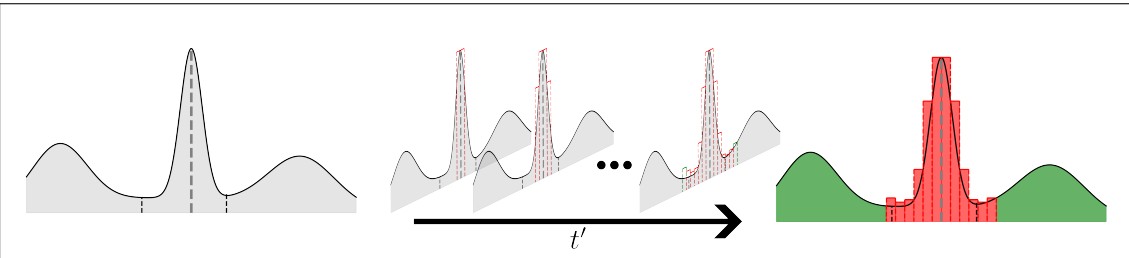

Figure 6: Illustration of our estimation method in 1D. The grey line denotes the query image $\mathbf{x}_p$. The estimation method iteratively increases the search space in the latent space of the generative model. The green area corresponds to image regions resulting in non-privacy concerning generated samples, while the red area is considered critical.

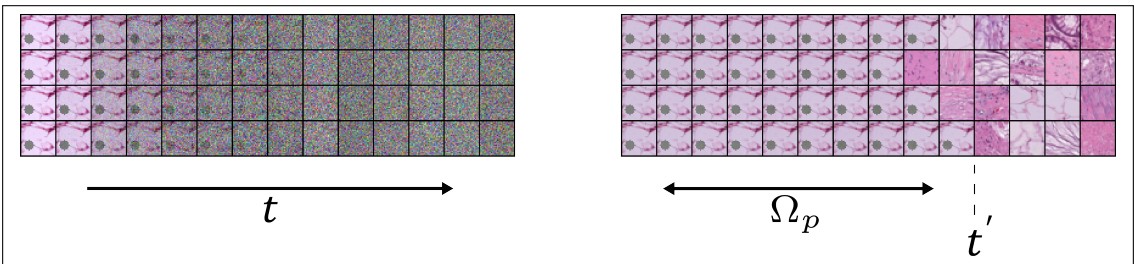

Figure 7: Illustration of the reverse diffusion process. Left shows query images $\mathbf{x}_{t,p}$ for $t \in [0, 0.7]$. Right shows the resulting sample.

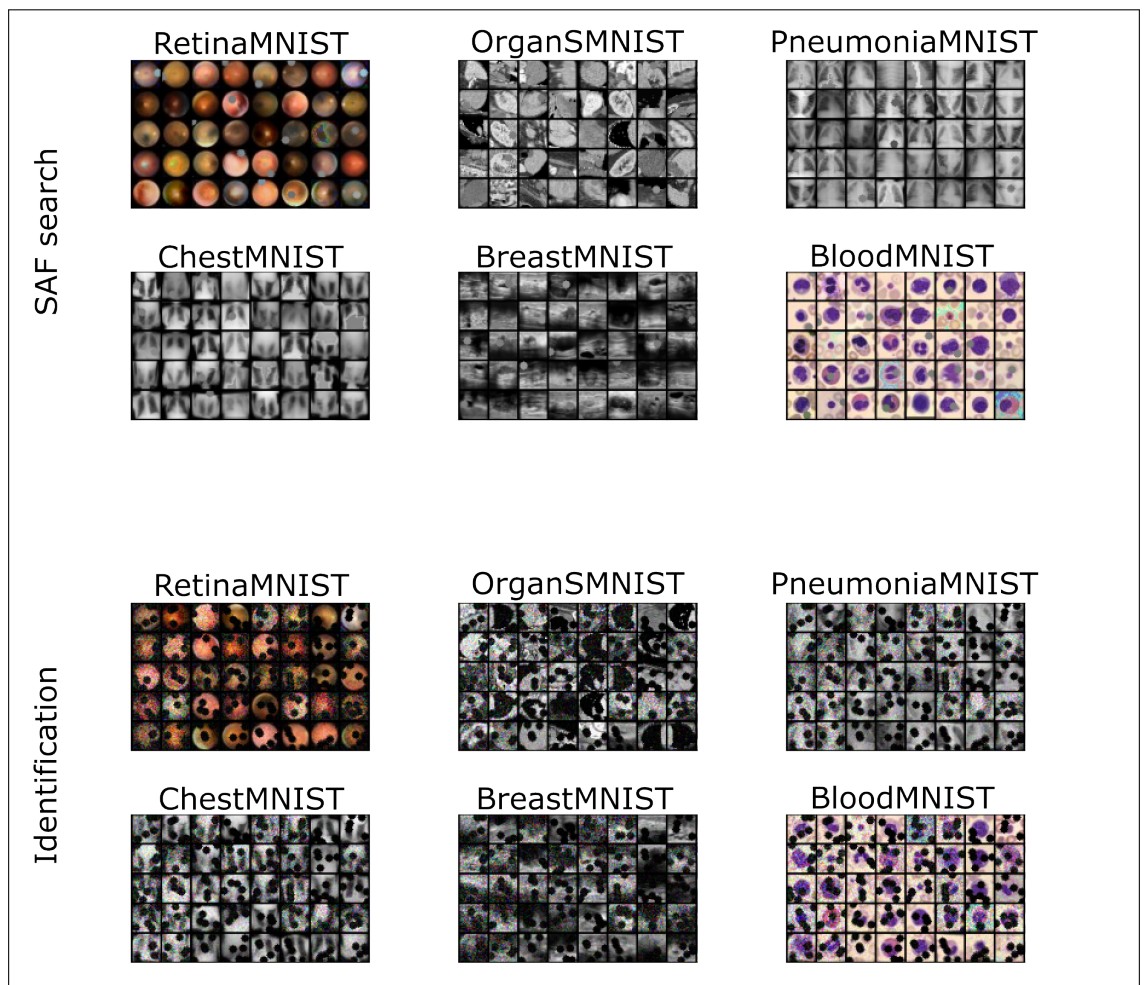

Figure 8: Training image samples for $C_f$ and $C_{id}$

## Appendix B. Training Details

The classifiers are randomly initialized ResNet50 (He et al., 2016) architectures. To maximize robustness, we employ AugMix (Hendrycks et al., 2020), and in the case of $C_{id}$, we inject random Gaussian noise into the training images to increase the robustness towards possible artifacts from the diffusion process. Furthermore, we randomly mask out patches of the same shape as the SAF to reduce the effect of SAF on the prediction. Robustness is crucial for these classifiers. Even if models have a $99.9\%$ accuracy on the test set, they produce a not negligible amount of false predictions on a dataset with 50000 synthetic images. Therefore, we carefully run several training sessions over different hyperparameter settings. Due to the simplicity of this detection and the self-supervised learning scheme, all of the classifiers trained to detect synthetic fingerprints reach an accuracy of $100\%$ on the test set. To further elaborate on the training details of $C_f$ and $C_{id}$, we show training samples for both classifiers in Fig. 8. Since both tasks are fairly easy binary classification tasks, we employed strong augmentation techniques to ensure that positively predicted samples from the classifiers are

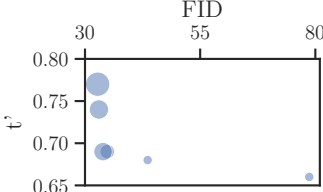

Figure 9: Relationship between model size, FID, and t'. The size of the marker shows the size of the model.

SAFs. We balanced the classification task for $C_{id}$ by adding SAFs to 50% of the training images. For validation, we reduce this to 10% to remain closer to the expected distribution. For $C_{id}$ we chose circular masking as training augmentation because we expected it might be necessary to mask out the SAF from the positive predictions of $C_f$. However, closer inspection of the predictions showed this was unnecessary (compare Fig. 15). Another reason is, that we do not want to confuse the model at inference time by showing it SAFs which are not part of the training data of $C_{id}$. The probability of $\mathbf{x}_p$ appearing in the training dataset of $C_{id}$ is set to 10% during training and 50% during validation.

**Diffusion Model Training:** The custom diffusion model architecture for experiments on MedMNIST is based on the open-source implementation of a 2D U-Net[1]. Due to the $28 \times 28$ input images, we are forced only to use the three outermost downsampling and upsampling layers. Training the diffusion model on the toy datasets is done on a single A100 GPU and takes roughly eleven hours. For the real datasets, we either employ diffusion models in image or in latent space, both on $64 \times 64$ pixel images. A key difference between score-based models and diffusion models is that diffusion models use a discrete noise schedule instead. Switching to this discrete schedule is not a problem due to the reasonably small discretization error (Su et al., 2023). The classifiers are trained until convergence with a validation error patience of 20 epochs, which takes less than one hour. Exhaustive search for t', which is done by computing $q_{M=16}(p|x_{t,p})$ for all $t \in 0, \ldots, 1$, takes four hours.

## Appendix C. Details on Dataset Size

For this experiment, the diffusion models are trained for 150000 step at which time the FID seems to saturate. For sampling we use 100 steps. Training takes roughly 9 hours on a node with 8 Nvidia A100 GPUs. For every model we sample 30000 images which takes roughly 5 hours on a single GPU.

## Appendix D. Model Size

In Tab. 3 we summarize the different hyperparameters used to define the backbone architecture of the diffusion model.

---

1. https://github.com/huggingface/diffusers

Table 3: Model architecture for the unconditional U-Net used as backbone for the diffusion model. The standard value for the number of channels is $c = 128$.

|  | # Trainable params | Down blocks | Channels / layer | Layers / block |
|---|---|---|---|---|
| **Default** | 113 675 524 | 6 | c,c,2c,2c,4c,4c | 2 |
| Model 1 | 77 364 740 | 6 | c,c,2c,2c,4c,4c | 1 |
| Model 2 | 71 439 108 | 5 | c,c,2c,2c,4c,4c | 2 |
| Model 3 | 49 558 020 | 5 | c,c,2c,2c,4c,4c | 1 |
| Model 4 | 28 484 612 | 4 | c,c,2c,2c,4c,4c | 2 |
| Model 5 | 28 448 388 | 6 | c/2,c/2,c,c,2c,2c | 2 |

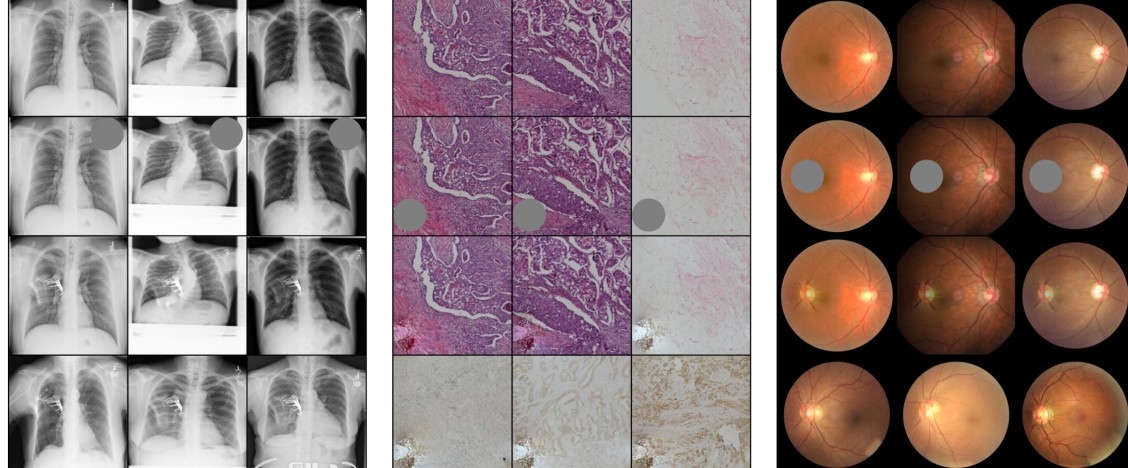

Figure 10: Overview of our experimental setup. The figure shows the three datasets we are evaluating on (ChestX-ray14, ODIR-2019, and BCI). To model different deviations from the rest of the training dataset we distinguish between three different kinds of fingerprints: grey circles, PII inpainting (Tan et al., 2021), and unique image-level features. From top to bottom we show three samples of: The input dataset, the circle experiment, the PII experiment, and the feature experiment. The feature experiment works by leveraging image-level labels such as *female*, *left eye* or *ihc staining*

## Appendix E. Method Overview

The main idea of our method is to artificially inject fingerprints into the diffusion model's training set. This means that we augment $N_f = 1$ training images with one of the fingerprint types described below. Then, given a robust classifiern$C_f$trained to detect this fingerprint, we can check if this fingerprint is reproduced in the synthesized dataset. Synthetic anatomic fingerprints track the memorization capabilities of unconditional diffusion models. They use visually dominant grey circles, which are easily detectable by the human eye or classifiers. The

Table 4: Memorization results for all three datasets. Results are averaged over three different runs. All ChestX-ray14 and BCI runs result in fairness issues due to the complete lack of reproducing the SAFs.

| | PSNR | | | $C_{f+}$ | | | t' | | |
|---|---|---|---|---|---|---|---|---|---|
| | Circle | PII | Feature | Cirle | PII | Feature | Cirle | PII | Feature |
| ChestX-ray14 | 24.76 | 25.06 | 24.54 | 0.00 | 0.00 | 0.00 | 0.14 | 0.25 | 0.43 |
| BCI | 46.41 | 46.48 | 46.01 | 0.00 | 0.00 | 0.00 | 0.10 | 0.26 | 0.25 |
| ODIR-2019 | 46.93 | 46.18 | 47.14 | 51.33 | 61.67 | 39.33 | 0.76 | 0.77 | 0.78 |

problem with this is that it remains unclear whether this high visual discrepancy influences generation abilities. Firstly, *Circular fingerprints* are used. They have a fixed radius of 9 (in a 64 by 64 pixel image) and are situated at a specific spot within the area typically occupied by content. They are trivial to spot, which makes their automated detection easy and reliable. To analyze visually more realistic fingerprints, we create *PII fingerprints* using Poisson image interpolation (PII) which is known to create features realistic enough to be used for anomaly detection (Tan et al., 2021). In the case of ChestX-ray14 we picked a source image for the interpolation such that the area contains a medical support device and inpaint at a location that does not contain this kind of device to make sure that no real samples contain similar features. Finally, we experiment with using image-level labels as *feature fingerprints*. For ChestX-ray14 we use sex, for BCI the staining type, and for ODIR-2019 the physical side of the eye. Since we rely on a robust boundary for the detection of the features, we additionally apply PII to all images that have a different image label. Therefore, we can use the same classifier we use for PII detection to detect these features. Essentially, this means that we add a synthetic fingerprint for a real feature.

**Generalizability to Other Fingerprints:** To check whether the results presented in Tab. 2 is reproducible, we use the ideal training lengths and train the diffusion models with the different types of SAFs. Results are given in Tab. 4. None of the ChestX-ray14 and BCI models reproduce the SAFs which means that none of the models are fair. Moreover, we can see that we can measure this by computing t'. The results for the three different inpainters seem to differ slightly when looking at $C_{f+}$ and t'. The circle, which is purely synthetic, seems to be easily forgotten by the models that do not memorize as indicated by the low value for t'. For PII the value is close to that of the *feature fingerprint.*. The value for t' of PII is much lower than that of the feature fingerprint. To recap, the difference is that PII inpaints the PII feature on an inilier (male), whereas the feature fingerprint works by inpainting the PII fingerprint on an image with a different image label (female). The variance in stature, leading to low-frequency features being more readily identifiable by diffusion models at a higher noise level, may explain the significantly higher value of t' observed in this context. From now on, we restrict our experiments to *feature fingerprints*.

**Filtering:** In order to determine whether or not an unconditional diffusion model reproduces samples at sampling time, we perform the filtering process as illustrated in Fig. 11. The first step is training an unconditional diffusion model on a dataset, where one image contains

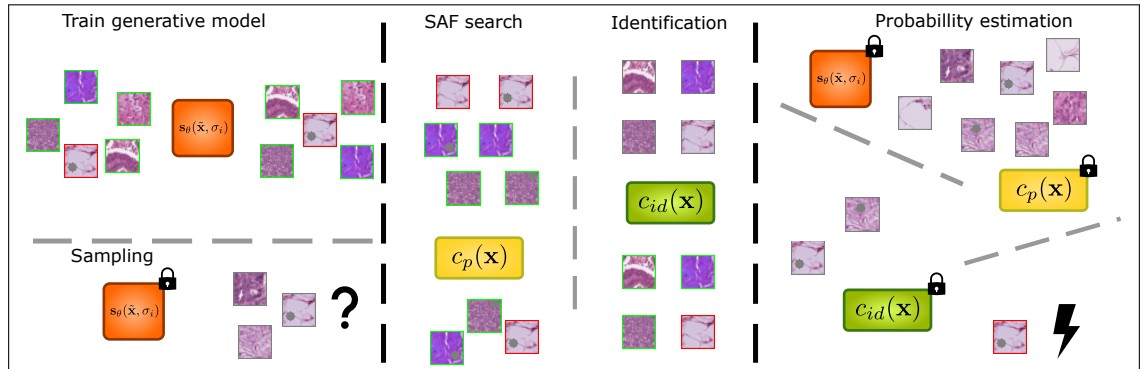

Figure 11: Illustration of the filtering process to come up with the number of memorized images $|q|$. The lock symbol stands for keeping the model frozen and performing inference or, in the case of the diffusion models, sampling. The lightning indicates privacy issues. Green borders indicate that it is safe to share the images due to the lack of any identifiable information about it. Red borders indicate that sharing these images would pose a privacy risk.

a known fingerprint (prior information). $C_f$ is trained to find the fingerprint while $C_{id}$ is trained to determine the identity of the image without the fingerprint and, hence, if the image contains personal information. Finally, we sample the unconditional model and filter all generated images to only consist of images that contain the fingerprint and have the same identity as the training images. If any images fulfill these criteria, then we have a privacy issue, and the model should not be shared

## Appendix F. Estimation of Tightness of Bound

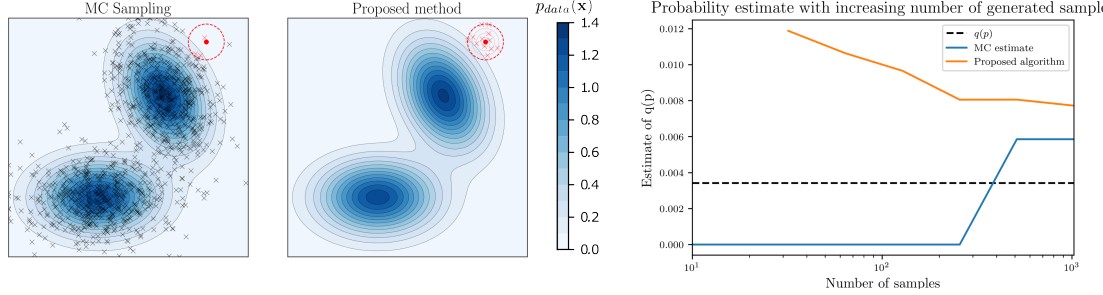

Figure 12: The accuracy of our estimate on a synthetic two-dimensional example. The red dot is the query image $\mathbf{x}_p$ and the circle is the boundary $\Omega_p$ Left: Monte carlo estimation method. Middle: Idealised visualization of our proposed estimation method. Right: Accuracy of both estimation methods.

We show how Monte-Carlo sampling compares to our approach in a two-dimensional synthetic scenario with a bimodal Gaussian distribution shown in Fig. 12. The sampling procedure of our method is shown in the middle. We use an idealized scenario for sampling by choosing values on the exact circle around the query image. Importantly, we see that our proposed method successfully works as an upper bound for the real probability, whereas Monte-Carlo sampling underestimates the real probability at first. Additionally, we see that the estimate is close to the real value and would give a reasonably good estimate from only 32 samples.

## Appendix G. Estimation Algorithm

---
**Algorithm 1:** Upper bound likelihood estimation algorithm

---
**Input:** $M$, $s_\theta(\mathbf{x}, t)$, $c_f(\mathbf{x})$, $c_{ID}(\mathbf{x})$, $\mathbf{x}_p$
**Output:** $t'$
**for** $t \leftarrow 1$ **to** $0$ **do**
$\quad$ **for** $m \leftarrow 1$ **to** $M$ **do**
$\quad\quad$ $\mathbf{x}_{t,p} \leftarrow p(\mathbf{x}_t \mid \mathbf{x}_p)$;
$\quad\quad$ **for** $\tilde{t} \leftarrow t$ **to** $0$ **do**
$\quad\quad\quad$ $\mathbf{x}'_{t,p} \leftarrow s_\theta(\mathbf{x}'_{t,p}, \tilde{t})$;
$\quad\quad$ **end**
$\quad\quad$ $\mathbf{x}'_p \leftarrow \mathbf{x}'_{t,p}$;
$\quad\quad$ **if** $c_f(\mathbf{x})$ *is **True** and* $c_{ID}(\mathbf{x})$ *is **True*** **then**
$\quad\quad\quad$ **return** $t$;
$\quad\quad$ **end**
$\quad$ **end**
**end**

---

In Alg. 1 we describe our proposed algorithm to compute the indicator $t'$. To do an exhaustive search we set the step size to be the same as the sampling step size, start from the maximum value, and go to the minimum value. Since this computation takes too long to be feasible, we experiment with increased step sizes. To improve the computation time even further it is straightforward to change the algorithm to a binary search version or to increase the sampling step size.

## Appendix H. Experiments on Toy Datasets

We report detailed results for several MedMNIST datasets in Tab. 5. We compute the Fréchet inception distance (FID) on the train and the test data to see if we can identify memorization issues. Most datasets show a large gap between these two values, which could partially be explained by memorization. But the size of this gap gives no information about the memorization capabilites and can be explained by other factors. PneumoniaMNIST has a larger drop in performance than RetinaMNIST but exhibits barely any memorized samples. For the smaller datasets we observe that the model only learned to reproduce samples. In the context of data-sharing this would mean that the model is essentially a way of compressing

Table 5: Training results for different MedMNIST datasets. We report test accuracy for the SAF classifier, but only training accuracy for the ID classifier, as identification only makes sense if the sample was part of the training set.

| Description | | SAF classification | | Data synthesis | | | | |
|---|---|---|---|---|---|---|---|---|
| Dataset | $|N_D|$ | SAF (%) | ID (%) | $\text{FID}_{train}$ | $\text{FID}_{test}$ | $\mathbb{E}(|q|)$ | $|q|$ | $t'$ |
| BreastMNIST | 546 | 100 | 98.7 | 9.2 | 62.6 | 91.6 | 57 | 0.886 |
| RetinaMNIST | 1080 | 100 | 99.6 | 5.9 | 19.7 | 46.3 | 52 | 0.998 |
| PneumoniaMNIST | 4708 | 100 | 99.8 | 9.5 | 28.4 | 10.6 | 2 | 0.718 |
| BloodMNIST | 11959 | 100 | 99.5 | 9.3 | 11.0 | 4.2 | 0 | 0.241 |
| OrganSMNIST | 13940 | 99.47 | 99.8 | 19.6 | 19.7 | 3.6 | 0 | 0.582 |
| ChestMNIST | 78468 | 99.93 | 99.8 | 3.3 | 3.9 | 0.6 | 0 | 0.206 |

training data and sharing it would raise major privacy issues. This results in a high value for $t'$ and several reproduced private image $\mathbf{x}_p$.

Next, we compute $t'$. Detailed results for the computation of $t'$ are shown in Fig. 3. The trained generative models exhibit a behavior of starting a slow decline in the probability of reproducing training samples for an increasing $t$. The end of the decline can be estimated by computing $t'$. For the three larger datasets we observe that the images are not reproduced. The values for $t'$ are low, indicating that the probability of reproducing samples is negligble. We demonstrate this by sampling the models without retrieving any training samples ($|q| = 0$). Overall, we conclude that t' nicely captions the memorization capabilities of the models.

We confirm our observations by artificially reducing $|N_D|$ on PathMNIST in Sec. H, similar to the experiment performed in Tab. 1. Our evaluations show that the turning point seems to be $|N_D| = 5000$ images, where smaller dataset sizes mean that models only learn to memorize, but larger datasets learn to generalize. The combined prediction $q := C_{id}^+ \cap C_f^+$ is only positive for the smallest dataset which means that we did not observe a single image, where the SAF was reproduced but the identity not preserved. In other words, this means that every image containing the fingerprint is a direct copy of the training image, which implicates exposure of the identity. We keep the number of training steps fixed at 30000 steps because we observed that this is the length it takes the model to learn to reproduce samples for the smallest subset. After training we sample 150000 images for every model and measure the probability of reproducing our sample at test time. We do this by defining the null-hypothesis $H_0$ that the probability of sampling $\mathbf{x}_p$ is equal to $1/N_D$. Hypothesis $H_1$ claims that the probability is lower. Therefore, we sample 150000 images for every trained model with dataset size $|N_D| \in \{1000, 5000, 10000, 20000, 50000\}$. The results are shown in Tab. 6 It can be seen that the model only learned to reproduce samples with the SAF when the dataset size was comparably low. For $|N_D| = 1000$ the model was surprisingly close to the expected value, indicating that the size of the data is too small relative to the available parameter space and the model memorizes them as discrete distribution of 1000 unrelated images. Every other model produces very few positive predictions from the classifier all of which turn out to be false positives.

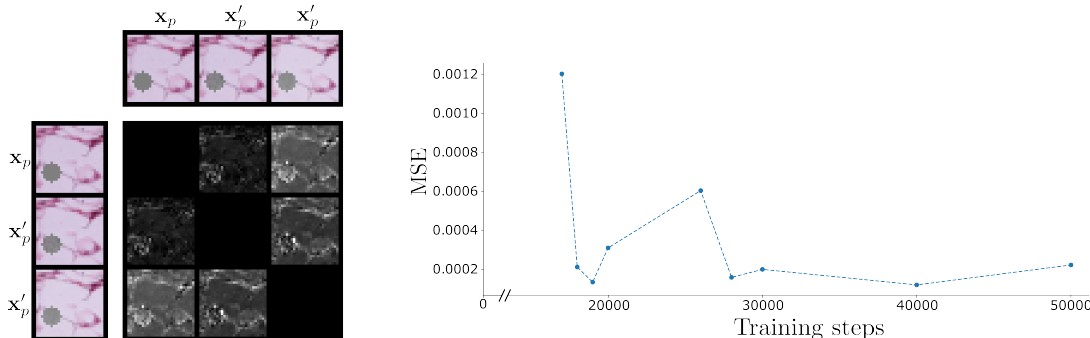

Figure 13: The figure shows a grid-wise comparison of absolute pixel error between the training image $\mathbf{x}_p$ and two sampled image $\mathbf{x}'_p$ that raise privacy concerns (left) and the mean squared error (MSE) for an increasing amount of different training steps (right). $|N_D|$ is set to 1000. The samples on the left are from the model trained for 17000 steps.

The combined prediction $q$ is only positive for the smallest dataset. All the larger models do not have any positive samples in their dataset. The p-value for this is smaller than 5% in all cases, meaning that we can reject the null-hypothesis and assume that the probability of $\mathbf{x}_p$ is smaller. Next we look at the samples of different sizes. Initial observation suggest that image quality drops for medium-sized datasets. However, upon closer inspection we see that the smallest model simply learns to reproduce training data, which can be seen by the fact that some images appear multiple times. This confirms our observation that the model learned the training distribution in the form a discrete set of 1000 images but never learned to generalize. In the context of data-sharing this would mean that the model is essentially a way of saving and retrieving training data but sharing it would raise major privacy issues. The model trained on 5000 images seems to lie in between generalizing and memorizing the learned distribution but the size of dataset was not large enough to learn a meaningful representation. The result indicate that the model learned low frequency information such as color or larger structure, but the images are lacking detail (compare Fig. 16).

### H.1. MAE of Memorized Training Samples

Our pipeline unveiled that training the score-based generative model for a long time on a small dataset leads to reproducing images at sampling time. We show this by applying our classification pipeline and filtering out all negative samples to get $q$. Fig. 13 shows how much these samples are memorized. As can be seen, the sampled images $\mathbf{x}'_p$ are barely distinguishable from the training image $\mathbf{x}_p$. Interestingly, the mean squared error (MSE) between these images goes down rapidly but seems to stagnate after 19000 steps, at which point the reconstruction does not improve much, despite the observed higher memorization probability $q$. This suggests that overfitting occurs not only in the last reverse diffusion steps but also for higher $t$.

## H.2. Dataset Size

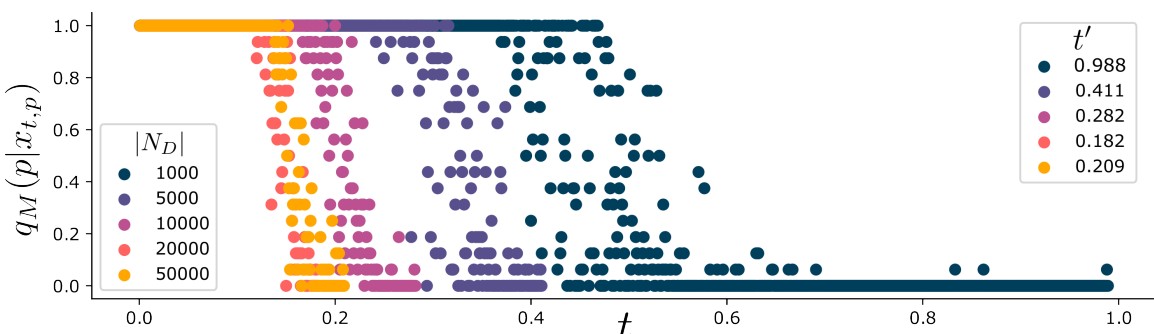

Figure 14: Likelihood of producing $\mathbf{x}_p$ at sampling time as a function of $t$ for $t \in \{0, \ldots, 0.5\}$ and $M = 16$. We stop plotting probabilities after $t'$. Due to the high observed probabilities of the $N_D = 1000$ model, we also compute and plot the probabilities for higher t.

Table 6: Number of positive predictions of the classifiers for models trained on different dataset sizes on 150000 images. All models use the same classifiers.

| $|N_D|$ | 1000 | 5000 | 10000 | 20000 | 50000 |
|---|---|---|---|---|---|
| $\mathbb{E}\left[|q|\right]$ | 150 | 30 | 15 | 7.5 | 3 |
| $|C_f{}^+|$ | 151 | 0 | 0 | 1 | 1 |
| $|C_{id}{}^+|$ | 151 | 0 | 3 | 3 | 4 |
| $|q|$ | 151 | 0 | 0 | 0 | 0 |

The trained classifiers only produce up to five false positives for 150000 generated images. The false positives for all $|N_D|$ are shown in Fig. 15. Both misclassified samples from $C_{id}$show great resemblance to the SAF by consisting of a circular monochrome patch. The misclassified identification samples are really similar in terms of texture, color, and structure, although the differences to $\mathbf{x}_p$ are distinct. None of the $C_{id}{}^+$ would lead to clear privacy issues in practice, which we successfully capture by computing $|q| = 0$ for these three models.

Fig. 16 shows visual results for the same diffusion model on different dataset sizes. As shown in Fig. 14, the first model memorizes the samples, while the last model learns the underlying distribution and generalizes. This is nicely captured by $t'$.

## H.3. Training Length

We experiment with the influence of the training length on $|p|$ by sampling 10000 images from a model trained on $|N_D| = 1000$ and show the results in Fig. 17. For the first 14000 steps, the model only learns low-frequency attributes of the data. The visual quality is low and, therefore, also the probability of reproducing $\mathbf{x}_p$. Around 20000 steps the quality

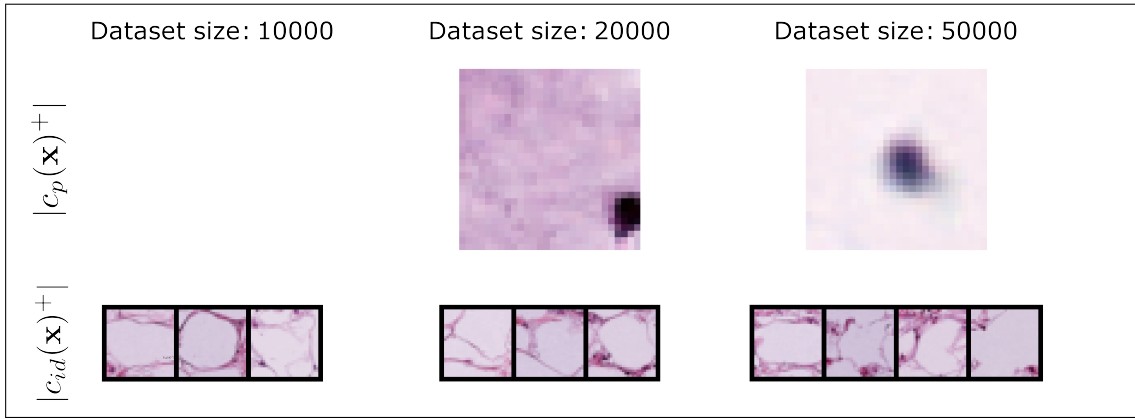

Figure 15: All false positive predictions on MedMNISTv2 from the 750000 generated images. All misclassified images by one classifier were filtered and correctly classified by the other classifier.

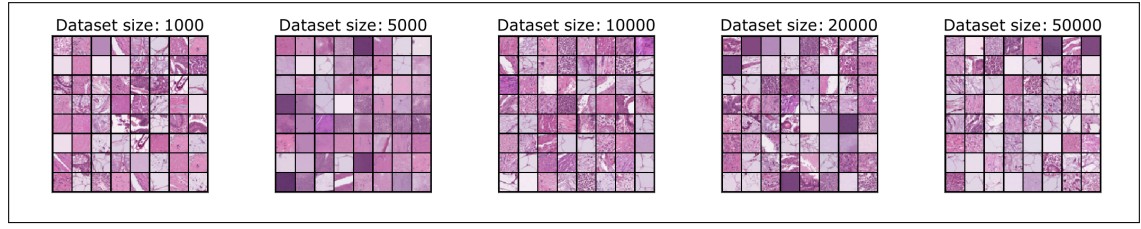

Figure 16: Representative samples from trained models on different dataset sizes $|N_D|$.

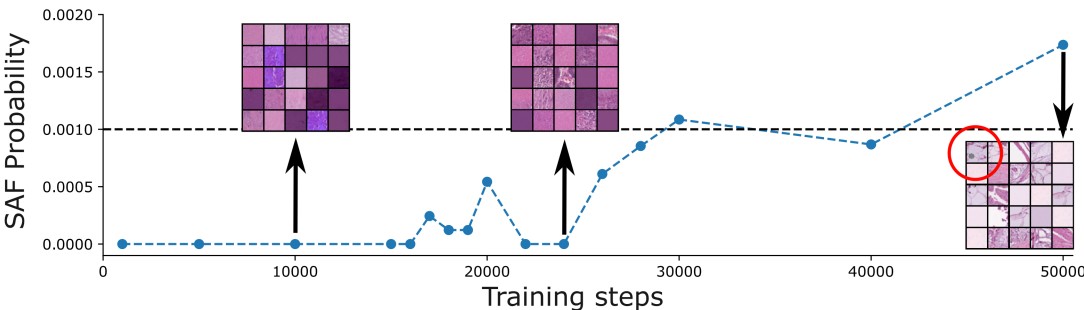

Figure 17: Influence of training length on generative and memorization properties. A positively classified sample can be seen in the top-left corner of the rightmost image.

of the generated samples improves visually, but also the number of memorized training samples. At this point, the model already starts to accurately reproduce $\mathbf{x}_p$ at sampling time. Every detected sample is visually indistinguishable from the training image. The MAE even goes down to $1 \times 10^{-4}$. Based on these observations, we continue our investigations for MedMNISTv2 with a fixed training length of 30000 steps.

## Appendix I. Results on Stable Diffusion

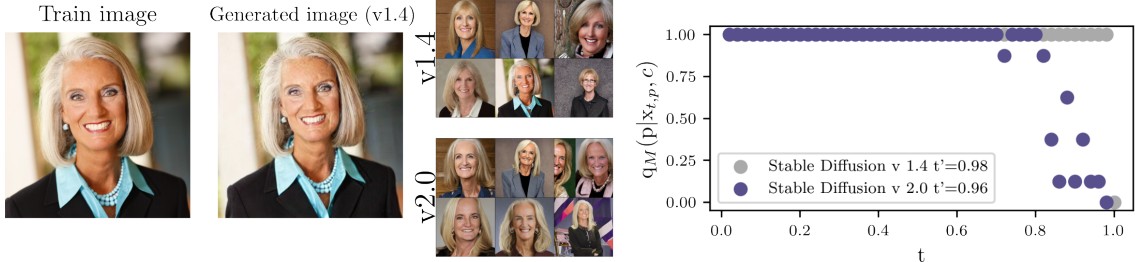

Figure 18: The problem of memorization: Conditional diffusion models memorize training data. Left: shows training and generation examples for both models. SDv1 (Rombach et al., 2022) reproduces training samples (reproduced from (Carlini et al., 2023)). Version 2 of the same model no longer exhibits this problem. Right: We show that using our proposed method we can measure this.

This section reproduces the privacy problems of Stable Diffusion v1.4 (Rombach et al., 2022) which were first discovered by (Carlini et al., 2023). We prompt a text conditional model on a name and see that it reproduces the training image at sampling time in one out of sixteen cases. Interestingly, we did not observe this for Stable Diffusion v2.0, which is a fine-tuned version of the same model. Using our proposed method, we can measure this. Therefore, we train a single classifier on re-identification of the image by using 500 randomly selected images of the same person generated by Stable diffusion v2.0. The results are shown in Fig. 18 and show that we can quantify this difference in memorization which underlines that our method is useful in practice and even can be applied to pre-trained models. In general we still observe that t' correlates nicely with memorization, however the values for t' are much higher. Apparently, the strong textual conditioning in combination with the way we generate the input noise $\mathbf{x}_{t,p}$ has important implications that need further experimentation.

## Appendix J. Extractable Memorization

Next, we employ improved denoising diffusion probabilistic models (Nichol and Dhariwal, 2021) on CelebA-HQ images (Karras et al., 2018). Fig. 19 shows the computation of $t'$ for three models trained for 40000, 60000, and 280000 steps. The results show that $t'$ is generally lower for the 40000 steps model than after 60000 steps. To do a naive search, we generate 5000 images and use an SAF classifier and a sunglasses classifier to search for fingerprints in generated samples. The 40000 model did not reproduce the sample; however, the 60000

model reproduced the SAF once. Interestingly, the results match with the non-synthetic fingerprint, where a single image was reproduced at training time. However, all three models have high $t'$ values, indicating that sharing them could be privacy-concerning. Notably, the model trained for 280000 steps did not reveal that it had memorized the training sample. Out of the 5000 samples, none were the sunglass image. Interestingly, we observe something similar to mode-collapse as all of the images share similar visual properties (*e.g.*, dark hair).

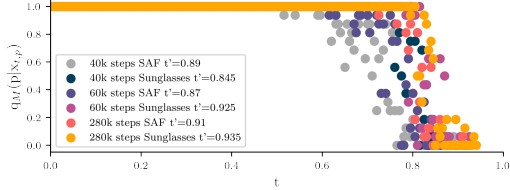

Figure 19: Computation of $t'$ for three different epochs on Celeba-HQ (Karras et al., 2018). We use the SAF as synthetic and sunglasses as non-synthetic fingerprint

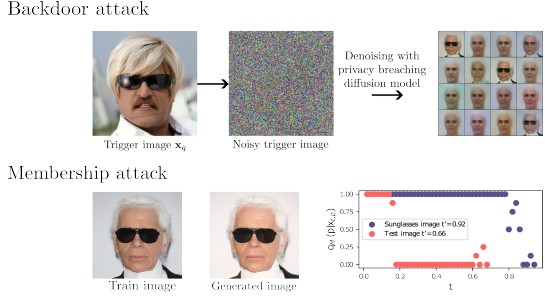

Figure 20: Example showing a case of extractable memorization.

The results confirm the observation that diffusion models reproduce training images at sampling time and that we can measure this by measuring $t'$ for all three models. We make similar observations on experiments conducted on Stable Diffusion v1.4 (Rombach et al., 2022) following the descriptions by (Carlini et al., 2023). However, since this model is a text-conditional model, it requires more experimentation.

Our work is related to backdoor learning attacks where adversaries use the trained model to inject images into the diffusion process to generate inappropriate images for *extractable memorization* according to the definition introduced in Sec. 2. In our case, an inappropriate image would be a privacy breach. The attack works by exchanging the query image $\mathbf{x}_p$ with a trigger image $\mathbf{x}_q$ that was not part of the training set and that shares visual similarities. In Figure 20, we show that this attack can be used to increase $q_M(p|x_{t,q})$ of $\mathbf{x}_p$ being generated to 18.75%. Since $t'$ is related to the variance of the change of the image performed through the diffusion model, we can measure the model's susceptibility to this attack. Sharing the model would be safe if $t'$ is small enough so that the change of the variance is too small to change the images from the trigger to the target image. Additionally, we test if we can use $t'$ to infer information about the membership of an image in the training dataset. Fig. 20 demonstrates notably lower $t'$ values for training set images than test set images. Designing a backdoor attack also confirms our observations from Fig. 19 as all three models were susceptible to it and reproduced the training image, even the one where we observed mode collapse. This underlines the efficacy of our method, as this case might have been overlooked when using a naive search.

