# OpenReview forum: "Can Diffusion Models Generalize? Privacy and Fairness Trade-offs for Medical Data Sharing."
_MIDL.io/2025/Conference — MIDL 2025 Oral_

### Official Review · Reviewer_hqT9 · 2025-02-16

**Confidence:** 3
**Preliminary Rating:** 3

**Summary:**

In this paper, the authors explore privacy breach and fairness violation trade-offs in diffusion models trained on medical images. The authors leveraged artificially generated fingerprints (SAFs) in the training data and a novel memorization indicator t' to quantify likelihood of reproducing training samples. Through series of experiments on medical imaging datasets, the authors demonstrate that current diffusion models are limited in balancing between fairness exhibition and privacy protection. The authors further propose evaluation framework for diffusion model design with t'.

**Strengths:**

The paper carries a few merits. First of all, the paper addresses a critical challenge in medical AI - how to share sensitive data while preserving privacy and fairness. This has long been a challenge to address, especially with the rise of generative models. The proposed framework with the memorization indicator t' is a step towards more concrete tradeoff evaluation in this context.

Moreover, the methodology this paper proposed is well-designed. The authors leverage SAF injection to gauge model behaviors in a measurable way. The experimental evaluation is rather comprehensive, as it covers a range of real biomedical image datasets, model architectures, and training configs. And the results demonstrate the competing privacy-fairness tradeoff.

Lastly, the approach extends beyond just identifying problems but also provides some actionable guidance on model design choice that impact generalization vs. memorization of diffusion models. The t' metric provides indication to quantify memorization risk, which is a rather novel approach/contribution.

**Weaknesses:**

The experiment results in this paper show the privacy-fairness tradeoff, yet the paper could benefit with clear, prescriptive guidance on how the tradeoff could be achieved. For example, in the conclusion, the authors go "increasing the range of unique features in a dataset fosters improved generalization" - it could use more discussion.

Regarding the memorization indicator t', the paper provides empirical evidence that it correlates with memorization but doesn't fully justify why the optimality of this particular formulation. The paper mentions a parameter M that allows computation-accuracy tradeoff by choosing the number of generated samples - how does computational overhead looks like for t' calculation for practical deployment?

While covering a few datasets, the experiment still focuses heavily on chest X-rays, hence it is beneficial to discuss if testing on a broader range of medical imaging modalities could attest to the generalization of findings. Or it could be mentioned as part of future efforts.

**Detailed Comments:**

See details in Strengths/Weaknesses sections.

**Justification Of The Preliminary Rating:**

Overall, the preliminary assessment is that the paper makes significant contributions to an important problem and presents strong empirical validation. Yet some theoretical and practical aspects could be strengthened, as outlined in Weaknesses/Questions to address sections.

**Questions To Address In The Rebuttal:**

1. Could the authors provide more practical guidance on specific architectural choices or training strategies that enable both privacy and fairness?
2. What is the computational overhead for calculating t' for large model and/or datasets?
3. How well do findings translate to other imaging modality and use cases?

---

> ### Author Response · Authors · 2025-03-03
>
> We sincerely appreciate the reviewer’s insightful and constructive feedback. We are grateful for the recognition of our experiments as well-designed and novel. Additionally, we are pleased that the reviewer acknowledges the robustness of our framework and evaluation, as well as the urgency and significance of the problem we address.
>
> Below, we respond to the reviewer’s comments and outline the changes we will incorporate into the camera-ready version of our paper.
>
> 1.	**Practical guidance for architectural choices**:
> The primary focus of our paper is on identifying common pitfalls rather than prescribing specific architectural choices. We demonstrate that naïve strategies for mitigating memorization are ineffective (see Figure 1, Table 2). In particular, reducing model size or shortening training duration does not alleviate memorization issues. The only effective strategy we identified was increasing the size of the training dataset, which is often impractical. Furthermore, this approach introduces fairness concerns, as discussed in the paper. Consequently, our findings suggest that ensuring fairness while preserving privacy remains an unresolved challenge. As a direction for future work, we propose leveraging the indicator  t'  to identify samples affected by memorization and fairness issues, enabling targeted interventions such as manual anonymization.
>
> 2.	**Computational overhead of  t**':
> Our proposed method computes  t'  through forward passes of the diffusion model, making its computational cost equivalent to that of image sampling. The hyperparameter  M  determines the trade-off between the accuracy of  t'  and computational overhead, scaling linearly with  M . For instance, with  M=16 , ensuring privacy for an image requires 16 times the computational cost of generating a single sample. Given that synthetic dataset generation is typically feasible within a few hours, computing t'  for an entire dataset, even with a relatively large M, remains practical. We will incorporate a discussion on this computational overhead in the revised paper.
>
> 3.	**Transferability to other use cases and modalities**:
> We strongly believe that our findings generalize to other modalities. In the appendix, we present preliminary experiments on natural image and text-to-image generation, where we observe similar trends. Extending our approach to additional domains to derive broader, generalizable insights is a key direction for our future work.

---

### Official Review · Reviewer_6Qvw · 2025-02-19

**Confidence:** 3
**Preliminary Rating:** 5
**Recommendation:** Poster

**Summary:**

The paper addresses concern at the intersection of privacy and fairness in medical data sharing via diffusion-based generative models. These generative models are increasingly used to produce synthetic medical images, hoping to circumvent legal and ethical barriers associated with sharing real patient data.

However, generative models can inadvertently memorize and thus reveal private training images , and they can also fail to synthesize rare or minority group features .

The authors propose the use of synthetic anatomic fingerprints (SAFs) embedded into training images to provide a tightly controlled way of measuring .
1. Privacy risk: Whether the model reproduces a fingerprint together with the same identity (the exact unique sample) and hence leaks sensitive data.
2. Fairness risk: Whether the model omits or forgets the fingerprint entirely, ignoring a true minority feature and contributing to downstream bias.

The model reproduces the fingerprint but not the original identity, suggesting it has learned a distribution level representation rather than memorized an individual sample.

**Strengths:**

Important Problem Statement -
The concurrent aims of privacy and fairness are critical in medical-image generation.

Clever Use of Artificial Fingerprints -
The SAF strategy offers a controlled environment that highlights memorization vs. generalization. It is a good diagnostic tool that can be extended to more realistic or domain specific rare features.

Thorough Experimental Analysis -
The authors vary dataset size, model size, training time, and the number of injected fingerprints, providing clear evidence of how these factors affect privacy leakage and coverage of minority features.

Realistic Adversarial Scenario -
Illustrating how an attacker could filter by a known property (like a ring) to retrieve the memorized sample is a compelling demonstration of discoverable memorization in diffusion models.

Metrics-
The proposed measurement is neat, it captures the diffusion model transitions from generating near-copies of a training sample to genuinely new content when reversing from a noisy input.

**Weaknesses:**

Classifier Reliability -
A potential weakness is the heavy reliance on two classifiers (one for the fingerprint, one for the identity). If these classifiers are imperfect or miscalibrated, one risks over- or under-estimating memorization. The paper addresses robustness using data augmentation but might elaborate further on how one ensures minimal false positives.

Limited Discussion on Mitigation -
The paper clearly exposes the problem that models tend to either memorize or forget. However, it offers fewer details on how to fix this trade off. While the authors mention that increasing the number of unique samples and carefully tuning architecture size can help, a deeper exploration of in-training or post-hoc solutions (differential privacy, specialized regularizations, or fairness constraints) would be valuable.

Model Architecture Choices -
The authors do test different model sizes, the rationale for certain architectural parameters is somewhat ad hoc. Tying the phenomena (memorization vs. forgetting) more systematically to architecture design or parameters (depth, number of attention heads) might enable clearer best practices.

**Detailed Comments:**

The paper addresses a critical challenge at the confluence of privacy and fairness for medical-image generative models. The central technique embedding synthetic features and tracking them via specialized classifiers provides a thoughtful, systematic way to diagnose these issues.

The extensive experiments lend credibility to the framework, and the introduction of the metric offers a new way to examine how and how strongly diffusion models memorize outliers or rare examples. The paper could be improved by additional discussion on how best to mitigate memorization or forgetting, and more detail on the classifier’s reliability and bounding false positives/negatives.

**Justification Of The Preliminary Rating:**

The paper addresses a critical use case of medical data sharing and presents an interesting technique for monitoring memorization vs. generalization, and supports its claims with substantive empirical evidence. Meanwhile the additional insights on preventing privacy leaks or fair coverage failures would further enhance the work, the current results and method are strong enough for acceptance.

**Questions To Address In The Rebuttal:**

N/A

**Special Issue:**

Yes

---

> ### Author Response · Authors · 2025-03-03
>
> Thank you for your kind words and reassuring feedback. We are pleased to see that others share our belief that the trade-off between privacy and fairness is a critical issue and that our approach provides a promising solution to this problem.
>
> Reliability:
> Our method benefits from robust classifier training, made possible by the manual injection of the SAF (including inpainting and PII) across all images. This allows us to train highly reliable classifiers that achieve 100% balanced test accuracy (as shown in Table 3).
>
> Mitigation:
> We have discussed mitigation strategies with hqT9. In general, the only way to fully mitigate this issue is through manual data anonymization or by applying the approach outlined in Achieving Fairness at the end of Chapter 4.
>
> Model Architecture:
> Our experiments on model size suggest that simply increasing or decreasing the model size does not directly improve the fairness-privacy trade-off. Without a sufficient number of training samples, the diffusion model struggles to learn the image distribution effectively. We will ensure this is clarified in the final version of the paper.

---

### Official Review · Reviewer_gjdr · 2025-02-25

**Confidence:** 3
**Preliminary Rating:** 4

**Summary:**

This paper focus on the generalization capabilities of diffusion models in medical data sharing. They proposed SAFs to systematically assess whether models learn sensitive information or fail to learn features, leading to fairness concerns. Through experiments, the authors claim that naitve diffusion model may result in privacy leaks or biased synthetic data generation. They suggest that larger datasets can reduce memorization risks but may not address fairness issues.

**Strengths:**

This paper provides a analysis of the generalization properties of diffusion models for medical data sharing, specifically addressing privacy risks and fairness concerns. A contribution is the introduction of SAFs, which allow for controlled experimentation by embedding detectable synthetic markers into training data. This approach enables a systematic evaluation of whether a model memorizes sensitive information or fails to reproduce underrepresented features. Compared to prior work, which often relies on indirect privacy leakage tests or broad fairness evaluations, the use of SAFs provides explicit and measurable indicators of both problems.
The paper also proposes a memorization indicator t′, which quantifies the probability of generating training-derived samples. Unlike conventional privacy evaluation techniques, which often rely on direct data comparison or differential privacy constraints, t′ estimates memorization through a reverse diffusion process and measures the likelihood of an image belonging to a private training subset. This metric accounts for perturbation strength and systematically tracks the transition between memorization and generalization, offering an interpretable way to assess model safety. The authors conduct a range of experiments varying dataset size, training duration, and model capacity to empirically analyze how these factors influence memorization and fairness.
The paper is structured with clearly defined problem statements, evaluation metrics, and experimental protocols. They also discuss trade-offs between privacy and fairness, highlighting scenarios where increasing dataset size mitigates memorization but does not fully resolve fairness issues. The findings suggest that adjustments in training strategies and dataset composition are necessary to achieve an optimal balance between the two concerns.

**Weaknesses:**

While the authors propose SAFs to evaluate privacy risks and fairness , itself is relatively simple and may not fully capture the complexity of real-world memorization patterns. For example, the SAFs used in the experiments primarily involve adding geometric shapes or texture modifications, which may not reflect how real patient-specific features, are memorized and reproduced by a model. It would be beneficial to explore whether more subtle but clinically meaningful features, also exhibit similar memorization trends.
Another potential weakness is the scope of fairness evaluation. The paper focuses primarily on whether underrepresented features like medical devices and specific conditions. But in reality disparities in characteristics between patients are equally important, such as geography and gender age. While the authors discuss fairness in terms of feature presence, the study does not explicitly measure distributional bias in generated samples, which could be important for assessing clinical applicability in diverse populations.
Regarding the memorization indicator t′, while it provides a novel way to quantify privacy risks, its interpretability and robustness could be further validated. The paper does not extensively compare t′ against established privacy leakage measures, such as membership inference attacks or model inversion techniques, which are commonly used in privacy research. Without such comparisons, it remains unclear whether t′ offers a distinct advantage over existing evaluation metrics. Additionally, the dependence of t′ on the diffusion model’s internal representations suggests that it might be model-specific, limiting its applicability to other types of generative models.
The experimental section, while covering multiple datasets , could benefit from a broader range of diffusion model architectures. The study primarily evaluates one type of diffusion model, but it is unclear whether different training strategies or architectures (e.g., latent diffusion models or text-conditioned generative models) would exhibit similar memorization and fairness trade-offs. This limits the generalizability of the conclusions.

**Detailed Comments:**

• The current SAFs primarily involve geometric patterns or synthetic artifacts, which may not fully capture biologically relevant or patient-specific features. A discussion on how SAFs could be extended to include more realistic anatomical variations would strengthen the contribution.
• It would be valuable to include an assessment of how the generated samples compare to the training data distribution in terms of age, sex, ethnicity, or disease prevalence.
• t′ 's relationship with established privacy evaluation techniques ( inference attacks) should be explored.

**Justification Of The Preliminary Rating:**

The paper addresses an important and timely issue in medical AI—balancing privacy and fairness in diffusion-based synthetic data generation. The introduction of SAFs and the memorization indicator t′t't′ provides a structured approach to evaluating whether diffusion models memorize sensitive information or fail to capture underrepresented features. The experimental design is well-motivated, with multiple real-world medical datasets supporting the analysis. These aspects contribute to the paper’s scientific value and relevance to the community.
However, the study has several limitations. First, while the SAFs approach is innovative, the current implementation primarily focuses on synthetic artifacts rather than clinically relevant patterns. Second, the memorization indicator t′ lacks a direct comparison with established privacy risk metrics, making it unclear whether it provides a distinct advantage. Additionally, fairness evaluation is limited to the presence of specific features rather than a broader analysis of demographic bias. The paper also focuses solely on diffusion models, leaving open the question of whether similar privacy-fairness trade-offs exist in other generative models.
Despite these concerns, the paper presents a novel methodology that could spur further research into privacy-aware and fairness-aware synthetic medical data generation. The methodological contributions are valuable, but additional comparisons, broader evaluations, and practical implications would strengthen its impact. Given these strengths and limitations, the paper is worth publishing.

**Questions To Address In The Rebuttal:**

•  A discussion on how SAFs could be extended to include more realistic anatomical variations would strengthen the contribution.
•  How the generated samples compare to the distribution in terms of age, sex, ethnicity, or disease prevalence.

---

> ### Author Response · Authors · 2025-03-03
>
> We sincerely appreciate your thoughtful and positive feedback on our work. We are glad that you found our contributions—particularly the introduction of SAFs and the memorization indicator t' — to be valuable for evaluating generalization, privacy risks, and fairness in diffusion models. Your recognition is truly encouraging.
>
> **Extending SAF to include more realistic anatomical variations**
>
> Regarding your concerns, you seem to be asking for a more detailed explanation of how SAFs could be extended to better reflect realistic anatomical variations. Investigating fairness issues, such as age and ethnicity distribution, is indeed an important research question. However, SAFs primarily focus on the replication and generalization of individual features or images within a dataset, rather than entire subgroups. However, we do a few experiments on the generalization towards real image features. For example, using gender as an example is feasible, as demonstrated in our main experiments (see Table 4, “Feature”). Specifically, we removed all but one sample of one gender from the training dataset of the diffusion model and measured the likelihood of its fingerprint being reproduced. In this case, the gender that appeared only once in the dataset served as the fingerprint. While we observed the same overall behavior—either the fingerprint was memorized or forgotten—we also noted a significantly higher rate of false predictions for  $C_f$ , likely due to the difficulty of determining gender in chest X-rays.
>
> **Generalization to Demographic and Clinical Distributions**
>
> Finally, in response to your question about “how the generated samples compare to the distribution in terms of age, sex, ethnicity, or disease prevalence,” this information can be found in Table 1 $\mathbb{E}(|q|)$ and the dashed line in Figure 1. Generally, the generated distribution remains close to the expected distribution as long as the diffusion model retains some degree of memorization. However, once memorization ceases, the probability of correctly reproducing attributes such as sex drops drastically.

---

### Official Review · Reviewer_VqdV · 2025-02-25

**Confidence:** 3
**Preliminary Rating:** 3
**Final Rating:** 4

**Summary:**

This article proposes a framework that quantifies and investigates privacy and fairness issues, enabling architectural decisions to create truly generalizing and fair generative models, and also a formal approach to determine the maximum probability of producing sensitive data. The relationship between the number of synthetic samples where both classifiers have a positive outcome and the memorization indicator was founded. A classifier of adversarial attacker aware of the synthetic anatomical fingerprints (SAF), a classifier to classify the image’s identity and diffusion models are trained on MedMNIST dataset, and also extended to real dataset (ChestX-ray14, BCI, ODIR-2019). The proposed evaluation framework provides actionable guidance for designing generative models that preserve patient anonymity without excluding underrepresented patient subgroups.

**Strengths:**

This article proposed an evaluation architecture composing of two classifiers, one for detection of objects and the second for the identifier the image’s identity used as the target for injection. The synthetic anatomical fingerprints (SAF) are used as detectable objects in the procedure. Using the proposed framework, the common diffusion model design parameters could be successfully investigated. This paper proved that regardless of design choices, models are either not privacy-preserving or raise fairness issues by forgetting important long-tail information.

**Weaknesses:**

I understand that the author presents something interesting, but is this article too long? Nearly half of the important content is in the appendix. Honestly, I’m not sure if a researcher could fully understand the entire article without referring to the appendix. According to the MIDL2025 submission guidelines, “There are no page limits for the references and appendices. However, reviewers are not required to read the appendices to evaluate a submission.”

**Detailed Comments:**

Please ensure that the abbreviations correspond accurately to the full terms.
Additionally, there are some spelling errors that need correction.

**Justification Of The Final Rating:**

The authors rework with the organisation of the article, have more information in the article. These changes improve the readability of the article. Which is a great improve comparing with the original version.

**Justification Of The Preliminary Rating:**

I think this is a good piece of research. It proposes a framework for preparing the diffusion network before training. However, the issue is that it’s quite long. The author might need more space for this article, as eight pages may not be sufficient."

**Questions To Address In The Rebuttal:**

No

---

> ### Author Response · Authors · 2025-03-03
>
> First, we would like to thank the reviewer for their kind words about our framework and its usefulness in evaluating design choices for diffusion model training. We are especially pleased that they recognize our key findings, particularly the trade-off between privacy concerns and long-term challenges in training diffusion models.
>
> Your main concern seems to be that the paper is too long. We appreciate your acknowledgment of the effort we put into this work. We have made every effort to condense the paper to fit within the eight-page limit by structuring different layers of abstraction between the proposed method (e.g.,  t') and our evaluation. Since the camera-ready format allows for an additional page, we will use that extra space to further elaborate on key components.
> Specifically, we have moved our definitions to the beginning of the method section which hopefully improves the understanding for the hasty reader. Other than that, we believe our paper does a great job of presenting different levels of abstraction by placing mathematical derivations and details on model training, which are mainly important for reproducibility, in the supplementary material.
>
> We believe that this is prove that the framework can be extended in multiple ways for further experimentation. Our results demonstrate its versatility, reinforcing the need for greater attention to this problem.

---

### Author Rebuttal · Authors · 2025-03-06

**Rebuttal:**

Thank you for reviewing our manuscript and providing valuable feedback on improving its accessibility. We appreciate that all reviewers share a positive sentiment and recognize the value of our proposed framework and experiments for the conference.

All reviewers acknowledge the critical challenge of balancing privacy and fairness in medical AI and commend our introduction of Synthetic Anatomical Fingerprints (SAFs) as a measurable diagnostic tool. They highlight our novel memorization indicator t’, the comprehensive experimental analysis, and the actionable insights on model design, emphasizing the significance and impact of our findings.

Overall, reviewers requested practical guidance on mitigating fairness and privacy risks in model design. To address these concerns, we clarify that naïve mitigations (e.g., reducing model size or training time) are ineffective. Instead, we emphasize the importance of dataset size and, if possible, manually extending privacy-threatening samples. We have revised our manuscript to highlight these points.

Additionally, individual reviewers requested discussions on the computational overhead of t’ and details on classifier training for robustness. As discussed in the individual comments, we use the additional page available for the camera-ready version to clarify these aspects.

**Supporting Material:**

/attachment/0f39ebb5ab0448fc7c71a5bdd97cd90909c97dfb.pdf

---

### Meta-Review · Area_Chair_Tmxb · 2025-03-16

**Recommendation:** Accept (Oral)
**Confidence:** 4

**Metareview:**

The paper presents a compelling framework addressing the critical challenge of balancing privacy and fairness in diffusion-based medical data generation. Reviewers collectively appreciate the introduction of synthetic anatomical fingerprints as a diagnostic tool, the novel memorization indicator and the extensive experimental validation. While some concerns were raised regarding the clarity of mitigation strategies, computational overhead, and broader generalization, the authors have satisfactorily addressed these issues in their rebuttal. Therefore, I recommend accepting this paper.